# A 33-mRNA Classifier Is Able to Produce Inflammopathic, Adaptive, and Coagulopathic Endotypes with Prognostic Significance: The Outcomes of Metabolic Resuscitation Using Ascorbic Acid, Thiamine, and Glucocorticoids in the Early Treatment of Sepsis (ORANGES) Trial

**DOI:** 10.3390/jpm11010009

**Published:** 2020-12-23

**Authors:** Jose Iglesias, Andrew V. Vassallo, Oliver Liesenfeld, Jerrold S. Levine, Vishal V. Patel, Jesse B. Sullivan, Joseph B. Cavanaugh, Yasmine Elbaga, Timothy E. Sweeney

**Affiliations:** 1Department of Critical Care, Department of Nephrology, Community Medical Center, Toms River, NJ 08755, USA; 2Department of Nephrology, Jersey Shore University Medical Center, Hackensack Meridian School of Medicine at Seton Hall Neptune, Nutley, NJ 07110, USA; 3Department of Pharmacy, Community Medical Center, Toms River, NJ 08755, USA; Vishal.Patel@rwjbh.org (V.V.P.); Joseph.Cavanaugh@rwjbh.org (J.B.C.); 4Inflammatix, Inc., Burlingame, CA 94010, USA; oliesenfeld@inflammatix.com (O.L.); tsweeney@inflammatix.com (T.E.S.); 5Department of Medicine Section of Nephrology, University of Illinois at Chicago, Chicago, IL 60612, USA; jslevine@uic.edu; 6Jesse Brown Veterans Affairs Medical Center, Chicago, IL 60612, USA; 7School of Pharmacy & Health Sciences, Fairleigh Dickinson University, Florham Park, NJ 07932, USA; jsull@fdu.edu; 8Department of Pharmacy, Monmouth Medical Center Southern Campus, Lakewood, NJ 08701, USA; Yasmine.Elbaga@rwjbh.org

**Keywords:** sepsis, septic shock, HAT therapy, vitamin c, endotyping, coagulopathic, hydrocortisone, thiamine, ascorbic acid

## Abstract

Background: Retrospective analysis of the transcriptomic host response in sepsis has demonstrated that sepsis can be separated into three endotypes—inflammatory (IE), adaptive (AE), and coagulopathic (CE), which have demonstrated prognostic significance. We undertook a prospective transcriptomic host response analysis in a subgroup of patients enrolled in the Outcomes of Metabolic Resuscitation Using Ascorbic Acid, Thiamine, and Glucocorticoids in the Early Treatment of Sepsis (ORANGES) trial. Methods: Blood was obtained from 51 patients and profiled using a pre-established 33-mRNA classifier to determine sepsis endotypes. Endotypes were compared to therapy subgroups and clinical outcomes. Results: We redemonstrated a statistically significant difference in mortality between IE, AE, and CE patients, with CE patients demonstrating the highest mortality (40%), and AE patients the lowest mortality (5%, *p* = 0.032). A higher CE score was a predictor of mortality; coronary artery disease (CAD) and elevated CE scores were associated with an increase in mortality (CAD: HR = 12.3, 95% CI 1.5–101; CE score: HR = 15.5 95% CI 1.15–211). Kaplan–Meier (KM) analysis of the entire cohort (*n* = 51) demonstrated a decrease survival in the CE group, *p* = 0.026. KM survival analysis of hydrocortisone, ascorbic acid, and thiamine (HAT) therapy and control patients not receiving steroids (*n* = 45) showed CE and IE was associated with a decrease in survival (*p* = 0.003); of interest, there was no difference in survival in CE patients after stratifying by HAT therapy (*p* = 0.18). These findings suggest a possible treatment effect of corticosteroids, HAT therapy, endotype, and outcome. Conclusion: This subset of patients from the ORANGES trial confirmed previous retrospective findings that a 33-mRNA classifier can group patients into IE, AE, and CE endotypes having prognostic significance. A novel finding of this study identifying an association between endotype and corticosteroid therapy warrants further study in support of future diagnostic use of the endotyping classifier.

## 1. Introduction

Sepsis, defined as a dysregulated immune response to an acute infection, confers an extremely high mortality and utilization of health care resources [1,2]. A complex interplay of host response and pathogen dynamics leads to varying outcomes and therapy responsiveness [3,4]. As a result, despite numerous clinical trials, anti-cytokine and targeted immune modulating therapies have failed to improve clinical trials in sepsis [1,5].

Recently, based on pre-clinical and clinical experience there has been great interest in employing the combination of hydrocortisone, intravenous ascorbic acid (AA) and thiamine (known as HAT therapy) in the management of patients with sepsis [6,7]. However, to date, studies have yielded conflicting results regarding the benefits of HAT therapy on clinical outcomes [6,8]. The landmark study by Marik, a propensity-adjusted observational study, reported a striking decrease in mortality in septic patients treated with HAT [9]. Fowler reported no difference in acute respiratory distress syndrome (ARDS), improved organ function, or inflammatory markers in patients receiving AA; however, there was a significant decrease in mortality [10]. The VITAMINS trial revealed no difference in clinical outcome in septic patients randomized to HAT therapy [11]. We recently conducted the double-blind placebo controlled trial Outcomes of Metabolic Resuscitation Using Ascorbic Acid, Thiamine, and Glucocorticoids in the Early Treatment of Sepsis (ORANGES), and reported that HAT therapy significantly reduced the time to resolution of shock [12]. Thus, identifying which (if any) patients will benefit from HAT remains a substantial clinical challenge.

Over the last decade microarray analysis, genomics and transcriptomics studies have demonstrated that the host response in sepsis explains some of the heterogeneity of the syndrome [3,4]. A promising new approach may be to identify immune subclasses (“endotypes”) based on host response transcriptomics as a way of directing immune-modulating therapies to those patients most likely to benefit [13].

Sweeney et al. previously reported the discovery and validation of three sepsis endotypes across 1300 patients with bacterial sepsis at hospital or ICU admission [14]. Whole blood transcriptomics identified three endotypes described as “Inflammopathic” (IE; high severity, high mortality, enriched for innate immune activation), “Adaptive” (AE; low severity, low mortality, enriched for adaptive immune activation), and “Coagulopathic” (CE; high severity, high mortality, possible disrupted coagulation) [14,15].

We here prospectively studied whether a previously defined 33-mRNA classifier for these transcriptomic endotypes holds predictive validity in septic patients in a substudy of the ORANGES trial [14].

## 2. Methods

Study population: Outcomes of Metabolic Resuscitation Using Ascorbic Acid, Thiamine, and Glucocorticoids in the Early Treatment of Sepsis (O.R.A.N.G.E.S.) ClinicalTrials.gov Identifier: NCT03422159.

The ORANGES trial was a randomized, double-blinded, placebo-controlled trial assessing the utilization of an ascorbic acid, thiamine, and hydrocortisone treatment bundle for the management of septic and septic shock patients admitted to an ICU and is described in detail elsewhere [12]. This study was performed from February 2018 to June 2019 in two community non-teaching hospitals in the United States. The study was approved by the Community Medical Center Institutional Review Board (IRB # 17-004) [12]. All participants were provided with written informed consent. For patients that presented with altered mental status or required mechanical ventilation, consent was obtained from the patient’s legally authorized representative. Patients were randomized to receive either ascorbic acid 1500 mg every 6 h, thiamine 200 mg every 12 h, and hydrocortisone 50 mg every 6 h or a matching saline placebo for a maximum of 4 days. Intensivists were allowed to order open-label corticosteroid therapy for patients as deemed necessary for their usual care (i.e., for respiratory failure). Prior to study therapy, initiation baseline ascorbic acid and thiamine levels were drawn and evaluated via liquid chromatography/mass spectrometry. Investigators were blinded up until patient enrollment ended and both primary and secondary study outcomes were met.

Sample collection: Whole blood was drawn in PAXgene Blood RNA tubes at enrollment along with other standard laboratory parameters. Data collection included demographic information, clinical scores (SOFA, APACHE II), laboratory results, length of stay, and clinical outcomes. Patients were followed up daily until time of discharge. PAXgene Blood RNA samples were shipped to Inflammatix (Burlingame, CA, United States), where RNA was extracted and the 33 mRNAs were quantitated using NanoString nCounter, as described [14,16].

Endotype measurement: Endotypes were calculated as previously described [14]. Briefly, each of the 33 mRNAs is assigned to one of three groups, and we calculated the difference of geometric means of gene expression for each grouping. The groupings are Inflammopathic (IE): *ARG1, LCN2, LTF, OLFM4, HLA-DMB*; Adaptive (AE), *YKT6, PDE4B, TWISTNB, BTN2A2, ZBTB33, PSMB9, CAMK4, TMEM19, SLC12A7, TP53BP1, PLEKHO1, SLC25A22, FRS2, GADD45A, CD24, S100A12, STX1A*; Coagulopathic (CE), *KCNMB4, CRISP2, HTRA1, PPL, RHBDF2, ZCCHC4, YKT6, DDX6, SENP5, RAPGEF1, DTX2, RELB*. We then applied the previously defined multi-class logistic regression model to these three input gene expression scores, which yields a probability of endotype assignment (for each subject, the total probability [p(Inflammopathic) + p(Adaptive) + p(Coagulopathic)] sums to 1). Each sample is assigned an endotype according to the highest probability [14].

Previously, cohort analysis linked the Coagulopathic endotype to clinical coagulopathy [14]. However, as the cohort was not prospectively enrolled to study coagulopathy, there was high missingness in the coagulopathy variables, and no data were collected on D-dimers, fibrinogen levels, or thromboelastography. Therefore, we did not perform analysis of coagulopathy here.

Statistical analysis: Summary statistics were computed for survivors, non-survivors, endotypes, and treatment arms. For continuous variables, medians and interquartile ranges were determined. When the assumptions of normality were not met, nonparametric tests such as the Kruskal–Wallis rank sum test or Mann–Whitney U test were employed. Categorical values were compared with Pearson’s chi-squared test or Fischer’s Chi-squared test when indicated. Significance was set at a p-value of less than 0.05. Cox Proportional Hazards analysis was employed to evaluate variables found to be statistically significant for mortality on univariate analysis. Kaplan–Meier survival analysis was performed comparing endotype survival outcomes stratified according to corticosteroid administration and HAT therapy. Statistical analysis was performed using SPSS^®^ and R^®^ (IBM. Chicago, Il., R Foundation for Statistical Computing, Vienna, Austria).

## 3. Ethics Statement

All procedures performed in the study were in accordance with the 1964 Helsinki declaration and its later amendments. Patients’ data were kept confidential, and no patients’ identifiers were included in data files handled for the purposes of this study.

## 4. Results

We prospectively evaluated host response endotype in 51 consecutive adult patients within 12 h of hospital admission diagnosed with sepsis who were enrolled in the ORANGES trial and consented to have testing for endotype analysis (Table 1) [12,14]. There was a total of 23 patients who received hydrocortisone, ascorbic acid, and thiamine (HAT) therapy, and 28 patients in the control (comparator) group available for analysis. There were 6 patients in the comparator group who received open label corticosteroid therapy at the decision of the intensivist. A total of 23 (45%) went on to mechanical ventilation. Overall, 8 patients (15%) died. Patients progressing to mortality were older *p* = 0.011, had coronary artery disease [8 (19%) vs. 7 (88%) *p* < 0.001], were on mechanical ventilation [16 (37%) vs. 7 (88%), *p* = 0.016], and had higher APACHE IV scores, *p* = 0.017 (Table 1).

The previously described peripheral blood-based 33-mRNA endotype classifier (calculated from bloodwork drawn prior to randomization) was used to designate every patient’s host response endotype as one of IE (34%), AE (39%), or CE (27%) [14]. Their clinical characteristics are given in Table 2. We also redemonstrate classifier confidence in identifying AE patients, with some overlap between IE and CE (Appendix A).

In keeping with previous large cohorts, we demonstrate that endotype grouping was associated with outcome, with the AE demonstrating a significant decrease in mortality (5%) when compared to both the IE (10%) and the CE (40%), *p* = 0.032 [3,14]. Kaplan–Meier analysis demonstrated differences in survival outcomes among endotypes (log rank *p* = 0.026) (Figure 1). Mortality predicted by Apache IV was 29% in IE, 17% in AE and 30% in CE (*p* = 0.023), and showed negative correlation with AE scores (R = −0.36, *p* = 0.011) and positive correlation with CE scores (R = 0.40, *p* = 0.005) (Figure 2)

Unlike previous studies, no difference in age was observed among endotypes in our cohort. The current study shows procalcitonin (not previously studied with sepsis host response endotypes) was highly induced in IE patients, but not in the AE patients (Figure 3). Similarly, the percentage of patients with positive blood cultures was highest in the IE patients (58%) as compared to AE (15%) and CE (15%) patients (*p* = 0.007).

In this cohort, several factors were significantly associated with mortality, including age, clinical severity (APACHE IV score), mechanical ventilation, the presence of coronary artery disease and the CE score. We employed Cox proportional hazards analysis and showed both coronary artery disease and CE score to be significant predictors of mortality (Hazard ratio (HR) 12.3, 95% CI 1.4–101, *p* = 0.020 & HR 15.5, 95% CI 1.14–211, *p* = 0.039, respectively).

We sought to evaluate the relationship between host response endotypes and therapy in the ORANGES study. The following groups were evaluated: all patients, those randomized to HAT (*n* = 23), a comparator group who received steroids off protocol (*n* = 6), and a comparator group who did not receive steroids per protocol (*n* = 22). In general, there were few differences between the groups (Table 3). Kaplan–Meier (KM) survival curves evaluating all subjects revealed a decrease in survival in CE patients (*p* = 0.026, Figure 1). This effect was also evident in the control group who did not receive corticosteroids (*n* = 22) (*p* = 0.002, Figure 4). Similar results were evident in the HAT therapy group and the comparator group who did not receive corticosteroids (*n* = 45) (*p* = 0.003, Figure 5A). However, evaluation of KM survival curves in the HAT therapy group showed the decrease in CE group survival previously noted was attenuated (*p* = 0.18, Figure 5B), suggesting a possible differential effect of corticosteroid and/or HAT therapy on the CE group. Another observation involving the KM survival curves analyzing HAT therapy and the control subjects not receiving corticosteroids (Figure 5A) we noted a small statistically significant decrease survival in the IE subjects. This decrease in survival in the IE group was further accentuated in the KM survival curves of HAT therapy alone (Figure 5B); however, this did not reach statistical significance. Although the number of subjects is small these findings possibly suggest a negative treatment effect of HAT therapy in the IE group.

We sought to understand the interaction of endotype and therapy on mortality outcome. However, numbers were too small for regression testing, so we provided raw counts (Table 4). Notably, the mortality in the CE group not receiving steroids (60%) falls substantially across both groups that received steroids (29%). We emphasize that the study is small and these findings are hypothesis-generating.

## 5. Discussion

Here, we prospectively confirm that the previously reported 33-mRNA sepsis endotype signatures, discovered and validated from *n* = 1300 cases across 23 cohorts are able to distinguish host response IE, AE, and CE endotypes in individual patients diagnosed with sepsis on admission [3,14]. Further supporting the findings of the previous study, the current study demonstrates a significant association with increased mortality in IE and CE patients, and a decreased mortality in AE patients [14]. In the IE and CE groups, we observed higher severity-of-illness scores, lower lymphocyte counts and higher APACHE-IV-predicted mortality. IE patients demonstrated higher levels of procalcitonin and more positive blood cultures [14]. CE patients demonstrated an increase in mortality in the setting of lower procalcitonin levels. With the exception of platelet count and INR, we lacked any specific measures of coagulopathy that would have enabled us to study a link between clinical coagulopathy and CE.

Comparing the cohorts from Sweeney et al. and the current study there are some differences. The current study was performed in community hospitals with a majority of patients being older and Caucasian with mainly with bacterial sepsis, primary bacteremia, and severe pneumonia. This is largely similar to most cohorts in the prior manuscript. In addition to being the first fully prospective validation in a bacterial sepsis cohort, the present manuscript was conducted in the setting of an RCT for HAT therapy, allowing us to study the potential therapy-predictive effects of the endotypes.

A novel hypothesis-generating finding of the current study is the demonstration of an association between endotypes and the response to HAT and corticosteroid therapy. Overall Kaplan–Meier and regression analysis demonstrate that CE is associated with the highest mortality whether or not patients receive HAT or corticosteroid therapy; however, these data also suggest a survival benefit in the CE group when corticosteroids are given, but no apparent survival benefit when steroids are given in the IE or AE groups. We also observed a small yet statistically significant decrease in survival of IE subjects receiving HAT therapy suggesting a possible negative treatment effect of HAT therapy in the IE group. This suggests the possibility that categorizing patients according to endotype groups can potentially identify patients who will respond to corticosteroid and/or HAT therapy. Although due to the small sample size, this finding will need to be confirmed in larger cohorts.

The current study demonstrates that septic patients generate a heterogeneous host response which poses significant challenges in risk stratification, resource utilization, and clinical response to therapy. Transcriptomic characterization of different host responses when employed with traditional clinical evaluation of mortality risk and organ dysfunction such as SOFA and APACHE score may improve mortality risk stratification and response to therapeutic interventions in individual patients [13,14,16,17]. Due to our small study size, the present results linking CE to improved outcomes with steroid treatment are only hypothesis-generating. With further study, it is possible that our endotypes molecular classifier could identify and stratify patients with different pathophysiologies matched as a companion-diagnostic test to guide a precision medicine-based stratification, prognosis, and intervention [14,15].

Our study has some major limitations, notably a small sample size from two centers, a relatively racially homogeneous population and some missing clinical data from the coagulation laboratory parameters. On the other hand, we used a preset tool (33-mRNA classifier) and validated preset clinical findings previously shown to be associated with the endotypes, lending credence to the findings.

The 33-mRNA classifier is one possible way to reduce the clinical heterogeneity and inform therapeutic decisions. We suggest that future studies of immunomodulatory therapy in sepsis should at a minimum draw RNA-stabilized blood at the time of enrollment so that a prospective blinded endotypes analysis (using ours or other classifiers) can be performed. It is also important that future studies include specific markers of coagulopathy in order to have a more robust analysis of CE. It is only through such further blinded prospective endotype-based analysis that we will gain the confidence to apply the endotypes in interventional treatment randomization.

More work is needed to identify and confirm a companion-diagnostic approach to immunomodulatory therapy and sepsis. We emphasize that transcriptomic evaluation of septic patients is leading to a paradigm shift in risk stratification and may enable the identification of patients responding to different therapeutic approaches, thereby leading to a personalized approach to individual patients.

## Figures and Tables

**Figure 1 jpm-11-00009-f001:**
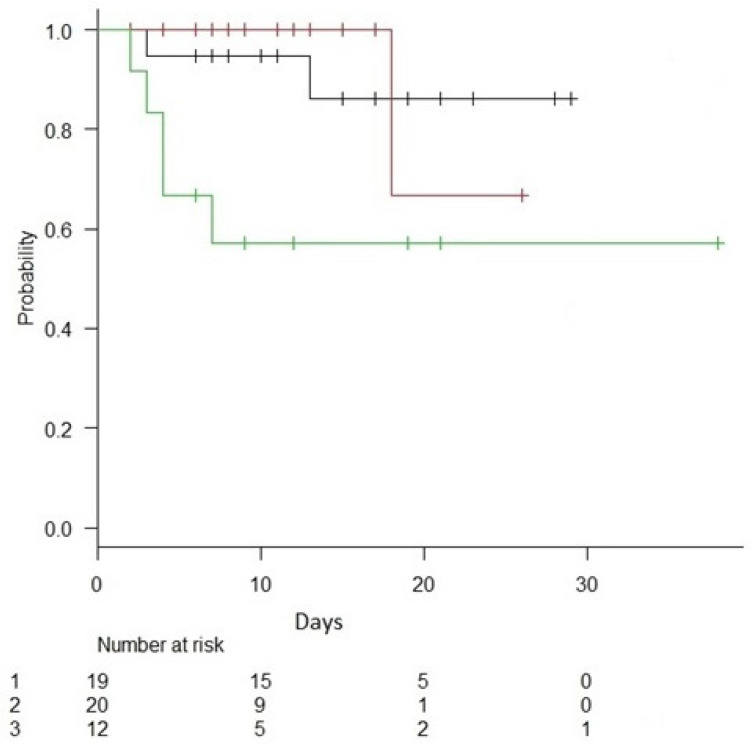
Kaplan–Meier survival curve across all endotypes (*n* = 51) IE (black line), AE (red line), CE (green line); (log rank with Bonferroni correction) (*p* = 0.026).

**Figure 2 jpm-11-00009-f002:**
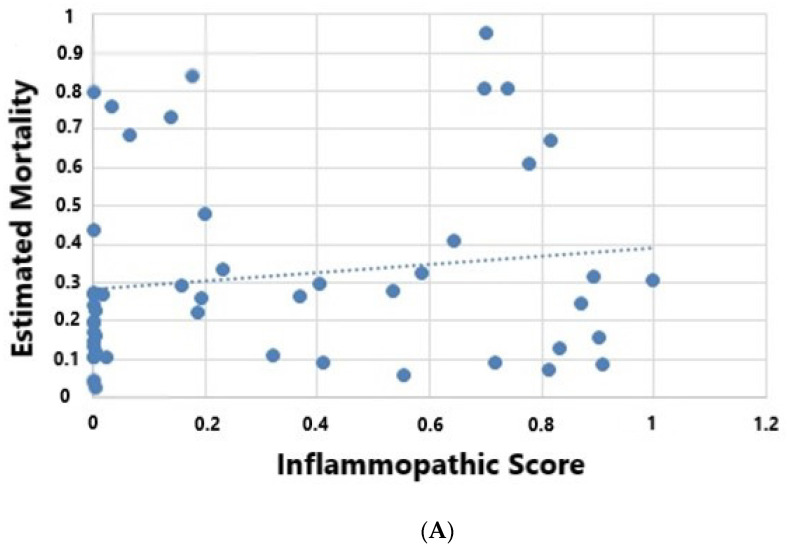
(**A**–**C**) Bivariate analysis (Pearson’s correlation demonstrating the relationship between sepsis host response endotype and Apache IV predicted mortality (Pearson’s correlation); (**A**), IE; R 0.19, *p* = 0.18, (**B**) AE; R-0.36, *p* = 0.011, (**C**) CE; R 0.40, *p* = 0.005.

**Figure 3 jpm-11-00009-f003:**
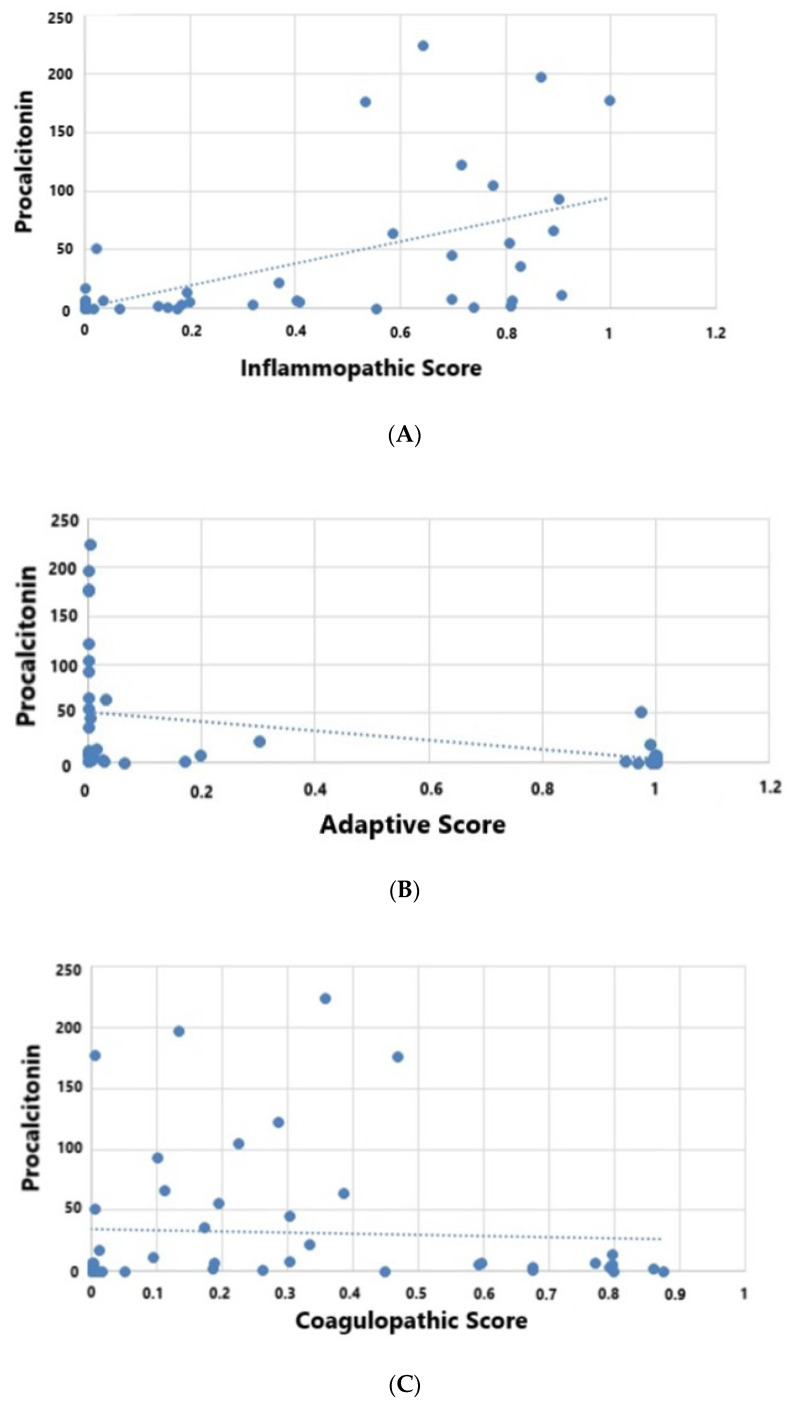
(**A**–**C**) Relationship between endotypes and procalcitonin levels (ng/mL). Bivariate Pearson’s correlation: (**A**) IE; R 0.63, *p* < 0.001, (**B**) AE; R 0.59, *p* < 0.001, (**C**). CE Correlation R = 0.22, *p* = 0.127.

**Figure 4 jpm-11-00009-f004:**
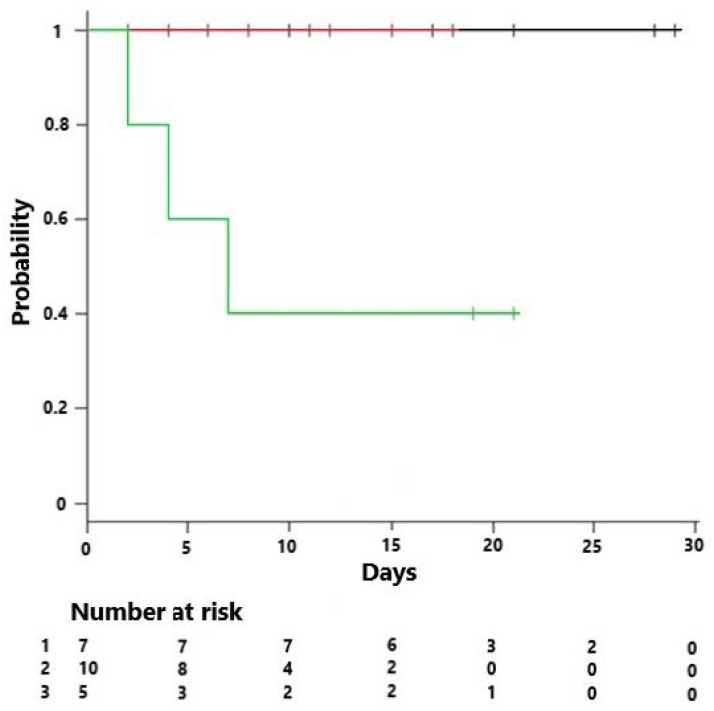
Kaplan–Meier survival curve in control subjects not receiving corticosteroids (*n* = 22) stratified according to endotype, IE (black line), AE (red line), and CE (green line) demonstrating decrease survival in the CE; log rank with Bonferroni correction (*p* = 0.002).

**Figure 5 jpm-11-00009-f005:**
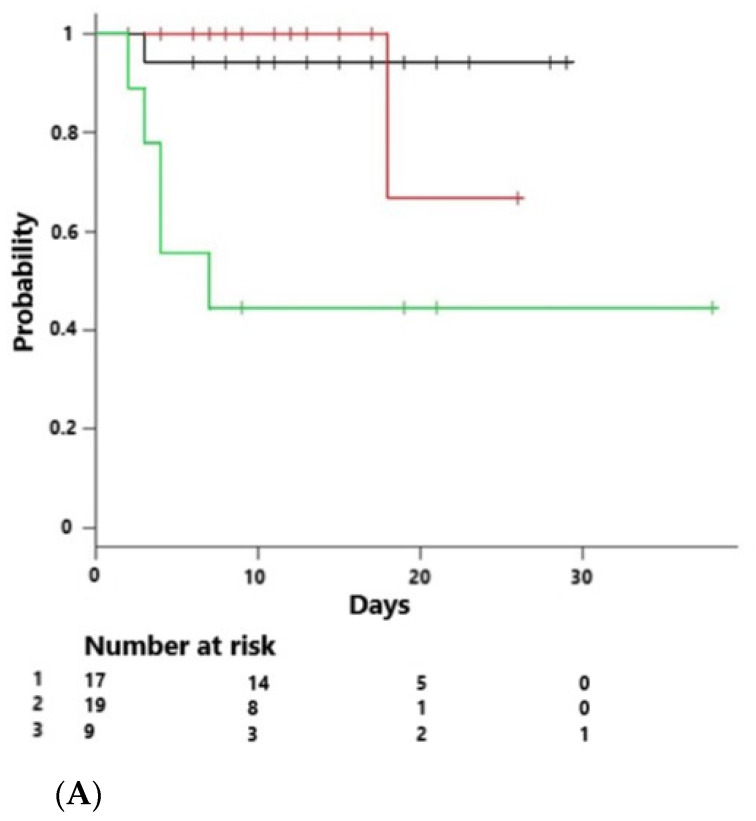
(**A**) Kaplan–Meier survival curve in HAT therapy group and comparator group who did not receive corticosteroids; IE (black line), AE (red line), CE (green line) (log rank with Bonferroni correction, *p* = 0.003). (**B**) Kaplan–Meier survival curve in HAT group (*n* = 23) stratified according to endotype, IE (black line), AE (red line), and CE (green line) (log rank with Bonferroni, *p* = 0.18).

**Table 1 jpm-11-00009-t001:** Baseline characteristics of survivors and non survivors.

	Survivors (*n* = 43)	Non-Survivors (*n* = 8)	*p*	OR	95% CI
Age	65 (59, 70)	79 (71, 86)	0.011		
Race (Caucasian)	41 (95%)	7 (88%)	0.41	0.34	0.027–4.29
Wt. (kg)	75 (66, 91)	76 (71, 96)	0.24		
Sex (male)	20 (47%)	3 (38%)	0.71	0.69	0.14–3.25
Diabetes	17 (40%)	4 (50%)	0.7	1.52	0.33–6.9
CHF	9 (21%)	1 (13%)	1	0.54	0.059–5.0
CAD	8(19%)	7(87%)	<0.001	30	3.2–265
COPD	14 (33%)	4 (50%)	0.43	2.07	0.45–9.52
CKD	7 (16.3%)	2 (25%)	0.62	1.71	0.28–10.3
Pneumonia	17 (40%)	3 (38%)	1	0.91	0.15–4.35
Primary bacteremia	7 (16%)	1 (13%)	1	0.73	0.073–6.94
Mechanical ventilation	16 (37%)	7 (88%)	0.016	11.2	1.33–105
Vasopressors	32 (74.4%)	5 (62%)	0.66	0.57	0.51–2.8
AKI	35 (81%)	7 (88%)	1	1.6	0.17–14.9
HD	6 (14%)	1 (13%)	1	0.88	0.091–8.4
Inflammatory	17 (39.5%)	2 (25%)	0.42	0.51	0.092–2.82
Inflammatory Score	0.18 (0.001, 0.69)	0.18 (0.08, 0.63)	0.42		
Coagulopathic	7 (16%)	5 (62%)	0.012	8.57	1.65–44.0
Coagulopathic Score	0.09 (0.003, 0.35)	0.77 (0.19, 0.84)	0.031		
Adaptive	19 (44%)	1 (13%)	0.12	0.2	0.020–1.65
Adaptive Score	0.17 (0.0001, 0.1)	0.01 (0.0007, 0.05)	0.16		
WBC × 10^9^/L	15 (11, 22)	15 (6, 22)	0.99		
Lactate (mM/L)	3.1 (2, 4)	4.9 (2.4, 9)	0.2		
SCr (mg/dL)	1.7 (1, 3.4)	1.5 (1.1, 2.1)	0.74		
Platelets (× 10^9^/L)	228 (168, 333)	192 (90, 334)	0.68		
Procalcitonin (ng/mL)	3.7 (0.8, 15)	1.65 (0.16, 8.5)	0.8		
Tbili (mg/dL)	0.7 (0.4, 1.6)	1.4 (0.6, 2.2)	0.34		
Ascorbic acid(mg/dL)	0.3 (0.1, 0.4)	0.52 (0.24, 0.67)	0.13		
Thiamine(nmol/L)	139 (113, 208)	118 (105, 138)	0.19		
SOFA	8 (5, 8)	9 (7.5, 10)	0.45		
APACHE II	21 (17, 28)	26 (22, 33)	0.12		
Apache IV	76 (66, 91)	107 (90, 119)	0.002		
Apache IV predicted mortality (%)	21 (11, 31)	64 (11, 31)	0.017		
PaO2/FiO2	256 (151, 341)	186 (95, 321)	0.58		
Total Lymphocyte	0.5 (0.3, 1)	0.35 (0.12, 0.7)	0.09		
INR	1.2 (1, 1.5)	2 (1.5, 2.3)	0.002		

Abbreviations/legend: OR = odds ratio, CI = confidence interval, CAD = coronary artery disease, COPD = chronic obstructive pulmonary disease, CKD = chronic kidney disease, CHF = congestive heart failure, AKI = Acute kidney injury, HD= hemodialysis, WBC = white blood cell count, tBili = total bilirubin, INR = international standardized ratio, PaO2/FiO2 = Partial pressure of oxygen/inspired concentration of oxygen ratio, SOFA = Sepsis-Related Organ Failure Assessment, APACHE = Acute Physiology and Chronic Health Evaluation.

**Table 2 jpm-11-00009-t002:** Baseline characteristics stratified according to endotypes at time of enrollment.

	Inflammatory (*n* = 19)	Adaptive (*n* = 20)	Coagulopathic (*n* = 12)	*p*
Age	64 (59, 73)	67 (54, 81)	76 (64, 82)	0.16
Race (Caucasian)	19 (100%)	18 (90%)	11 (84%)	0.53
Weight (kg)	70 (60, 84)	86 (63, 100)	76 (70, 100)	0.54
Sex (male)	10 (53%)	5 (25%)	8 (61%)	0.061
CAD	5 (26%)	4 (20%)	6 (46%)	0.18
Diabetes	4 (20%)	10 (50%)	7 (54%)	0.071
CHF	2 (10%)	5 (25%)	3 (23%)	0.28
COPD	8 (42%)	5 (25%)	5 (38%)	0.46
CKD	5 (26%)	4 (20%)	0 (0%)	0.16
Morbid obesity (BMI > 40)	2 (10%)	5 (25%)	1 (8%)	0.33
Pneumonia	8 (42%)	6 (30%)	6 (46%)	0.66
Primary bacteremia	6 (31%)	1 (5%)	1 (8%)	0.05
Mechanical ventilation	10 (53%)	5 (25%)	8 (61%)	0.051
Vasopressors	14 (74%)	13 (65%)	10 (77%)	0.52
AKI	15 (79%)	16 (80%)	11 (85%)	0.62
Positive blood cultures	11 (58%)	3 (15%)	2 (15%)	0.007
WBC × 10^9^/L	14 (5, 27)	14 (10, 20)	18 (15, 22)	0.12
Lactate (mM/L)	3.8 (2.8, 3.6)	2.7 (1.6, 3.6)	4.5 (2, 6.5)	0.19
SCr (mg/dL)	2 (1.3, 3.6)	1.36 (0.96, 2.1)	1.6 (1.2, 2.1)	0.21
Platelets (× 10^9^/L)	189 (93, 235)	263 (205, 370)	213 (167, 350)	0.024
Procalcitonin (ng/mL)	22 (4.6, 60)	0.8 (0.2, 3.6)	3.6 (1.6, 7)	<0.001
Tbili (mg/dL)	0.9 (0.6, 2)	0.5 (0.4, 0.7)	1.2 (0.62, 2)	0.006
PaO2/FiO2	211 (133, 293)	277 (166, 366)	224 (86, 347)	0.43
SOFA	9 (7, 13)	6 (4, 9)	8 (6, 10)	0.036
INR	1.28 (1.1, 1.5)	1.25 (0.96, 1.6)	1.4 (1, 2)	0.87
Total lymphocytes	0.4 (0.2, 0.6)	0.9 (0.4, 1.4)	0.4 (0.25, 0.8)	0.038
APACHE II	25 (18, 32)	20 (15, 30)	23 (20, 32)	0.079
Mortality across endotypes (%)	10%	5%	40%	0.032
APACHE IV	89 (65, 113)	74 (65, 79)	93 (73, 115)	0.053
Apache IV predicted mortality (%)	0.29 (0.12, 0.62)	0.17 (0.1, 0.26)	0.3 (0.23, 0.7)	0.023
Ascorbic acid (mg/dL)	0.3 (0.1, 0.4)	0.3 (0.12, 0.4)	0.4 (0.28, 0.65)	0.19
Thiamine(nmol/L)	187 (117, 253)	126 (104, 153)	139 (107, 175)	0.20

Continuous variables expressed in Median (IQR) Abbreviations/legend: OR = odds ratio, CI = confidence interval, AKI = acute kidney injury, CAD = coronary artery disease, COPD = chronic obstructive pulmonary disease, CKD = chronic kidney disease, WBC = white blood cell count, tBili = total bilirubin, INR = international normalized ratio, SCr = serum creatinine, PaO2/FiO2 Partial pressure of arterial oxygen/inspired concentration of oxygen.

**Table 3 jpm-11-00009-t003:** Patient characteristics stratified according to treatment.

	HAT Therapy (*n* = 23)	Control (+ Steroids) (*n* = 6)	Control (no Steroids) (*n* = 22)	*p*
Race (Caucasian)	22 (96%)	6 (100%)	20 (91%)	0.64
Age	64 (58, 80)	73 (64, 79)	67 (60, 77)	0.77
Sex (M)	13 (57%)	13 (50%)	7 (32%)	0.24
Weight	74 (60, 89)	105 (78, 121)	74 (63, 96)	0.076
Pneumonia	9 (39%)	5 (83%)	6 (27%)	0.045
Primary Bacteremia	3 (13%)	2 (33%)	3 (14%)	0.44
CAD	5 (21%)	1 (17%)	9 (41%)	0.28
CHF	4 (17%)	2 (33%)	4 (18%)	0.66
DM	6 (26%)	3 (50%)	12 (54%)	0.14
Mechanical ventilation	11 (48%)	4 (67%)	8 (30%)	0.38
Vasopressors	20 (87%)	4 (67%)	13 (59%)	0.1
ARF	12 (52%)	3 (50%)	7 (32%)	0.1
HD	3 (13%)	0 (0%)	4 (18%)	0.51
Inflammopathic	9 (37%)	1 (5%)	10 (50%)	0.46
Inflammatory Score	0.31 (0.0001, 0.73)	0.36 (0.002, 0.81)	0.15 (0.007, 0.6)	0.58
Coagulopathic	10 (43%)	2 (33%)	7 (32%)	0.73
Coagulopathic Score	0.13 (0.002, 0.35)	0.39 (0.13, 0.77)	0.074 (0.003, 0.50)	0.33
Adaptive	4 (17%)	3 (50%)	5 (13%)	0.57
Adaptive Score	0.009 (0.0001, 0.9)	0.035 (0.0001, 0.39)	0.09 (0.007, 0.9)	0.48
WBC (K/mL)	14 (7, 20)	18 (12, 23)	16 (12, 25)	0.29
Lactate (mM/L)	2.6 (1.9, 4.1)	4 (1.72, 4.5)	3.5 (2.9, 6.5)	0.29
Creatinine (mg/dL)	1.63 (1.26, 3.62)	1.7 (1.25, 2.01)	2.5 (0.74, 4.63)	0.46
Total Lympocyte	0.4 (0.2, 0.8)	0.8 (0.45, 1.6)	1.4 (1.06, 1.55)	0.096
Platelets (K/mL)	200 (85, 277)	185 (161, 273)	269 (203, 311)	0.2
INR	1.28 (1.1, 2.1)	1.04 (1.03, 1.05)	1.4 (1.06, 1.55)	0.23
Procalcitonin (ng/mL)	3 (0.19, 8)	7 (6.4, 22)	2.5 (0.74, 4.7)	0.1
Tbili (mg/dL)	0.8 (0.5, 1.7)	0.85 (0.47, 1.2)	0.7 (0.4, 1.85)	0.98
PaO2/FiO2	200 (85, 277)	185 (161, 273)	96 (69, 206)	0.06
SOFA	9 (7, 11)	8.5 (5.5, 10.7)	6 (4, 9)	0.16
APACHE II	23 (19, 30)	21 (15, 39)	21 (17, 30)	0.76
APACHE IV	82 (73, 91)	84 (57, 121)	70 (65, 97)	0.49
APACHE IV predicted mortality (%)	27 (17, 41)	34 (12, 69)	16 (10, 31)	0.26
Ascorbic acid (mg/dL)	0.3 (0.1, 0.4)	0.3 (0.12, 0.4)	0.45 (0.12, 0.67)	0.14
Thiamine (nmol/L)	187 (151, 153)	126 (104, 153)	140 (115, 178)	0.19

Continuous variables expressed in Median (IQR) Abbreviations/legend: OR = odds ratio, CI = confidence interval, AKI = acute kidney injury, CAD = coronary artery disease, COPD = chronic obstructive pulmonary disease, CKD = chronic kidney disease, WBC = white blood cell count, tBili = total bilirubin, INR = international normalized ratio, SCr = serum creatinine, PaO2/FiO2 = Partial pressure of arterial oxygen/inspired concentration of oxygen ratio, SOFA = Sepsis-Related Organ Failure Assessment, APACHE = Acute Physiology and Chronic Health Evaluation, (*p* values adjusted with Bonferroni correction).

**Table 4 jpm-11-00009-t004:** Stratification of mortality by endotype assignment and therapy group.

**HAT Therapy**	***n* = 10**	***n* = 9**	***n* = 4**
	IE	AE	CE
survival	9	8	2
mortality	1	1	2
**Controls given steroids off protocol**	***n* = 2**	***n* = 1**	***n* = 3**
	IE	AE	CE
survival	1	1	3
mortality	1	0	0
**Controls no steroids**	***n* = 7**	***n* = 10**	***n* = 5**
	IE	AE	CE
survival	7	10	2
mortality	0	0	3

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
