# Peer review of "A 33-mRNA Classifier Is Able to Produce Inflammopathic, Adaptive, and Coagulopathic Endotypes with Prognostic Significance: The Outcomes of Metabolic Resuscitation Using Ascorbic Acid, Thiamine, and Glucocorticoids in the Early Treatment of Sepsis (ORANGES) Trial"

_jpm, 2020, doi:10.3390/jpm11010009_

Round 1

Reviewer 1 Report

In this manuscript, Iglesias et al., aimed to validate the 33-mRNA classifier in separating sepsis endotypes by investigating a subset of patients in the ORANGES trial and examine the association between endotypes and mortality, or between endotypes and response to HAT and corticosteroid therapy.

In general, this is a prospective study for validating previously proposed one of the sepsis endotype classification based on transcriptomic host response. Overall, this manuscript is well written, and most of the statistical analysis is clear appropriate. The conclusion is promising for the use of these endotypes of sepsis in guiding personalized therapy. However, this reviewer has the following concerns on the current version of the manuscript.

Major:

  • The novelty of this study can be improved. The authors claimed that they repeated the previous finding by using 33-mRNA classifier for the endotype classification of sepsis. Indeed, this statement comprised their novelty. The authors should classify more detail of the differences of their study compared with the previous one [Ref 14], such as the kinds of sepsis when admitted to ICU, only have bacterial sepsis or more than that?
  • It was not clear how the authors choose this subgroup (51 patients) from the ORANGES trial for the current study? How the author determined the sample size and the randomization?
  • The authors mentioned that there are 6 patients in the control (comparator) group who received open label corticosteroid therapy at the decision of the intensivist. Thus, the control group was divided into two subgroups, named control with or without corticosteroid. They further combined the HAT group with these 6 patients as corticosteroid therapy group as in Table 4B. This reviewer does not think it is appropriate since this is actual different therapy group (3 medication vs 1).

  • In the recent report of ORANGES trial (n = 137) by the same group they authors concluded that No statistically significant change in SOFA score was found between groups, P = 0.17. No significant differences were found between study arms in ICU and hospital mortality, although they found significant difference in the time patients required vasopressors [ref 12]. However, in the current study, even in a subgroup from ORANGES, they authors found that HAT+CE can significantly eliminate the significant association of CE with mortality (Table 4A), even reduce in mortality in CE when all subjects were stratified by corticosteroid therapy (Table 4B). As the total CE patient is 12 shown in Table 2, the n of HAT within CE (HAT+CE) must be smaller. How could that reach to the above results?
  • Compare with Fig 5A and 5B, the IE survival curve seems even worse in HAT group alone vs HAT+comparator group without corticosteroids? Please confirm and discuss.

Author Response

We greatly appreciate the reviewers observations comments and recommendations.  Below are our responses.

Reviewer 1:

In this manuscript, Iglesias et al., aimed to validate the 33-mRNA classifier in separating sepsis endotypes by investigating a subset of patients in the ORANGES trial and examine the association between endotypes and mortality, or between endotypes and response to HAT and corticosteroid therapy.

In general, this is a prospective study for validating previously proposed one of the sepsis endotype classification based on transcriptomic host response. Overall, this manuscript is well written, and most of the statistical analysis is clear appropriate. The conclusion is promising for the use of these endotypes of sepsis in guiding personalized therapy. However, this reviewer has the following concerns on the current version of the manuscript.

Major:

The novelty of this study can be improved. The authors claimed that they repeated the previous finding by using 33-mRNA classifier for the endotype classification of sepsis. Indeed, this statement comprised their novelty. The authors should classify more detail of the differences of their study compared with the previous one [Ref 14], such as the kinds of sepsis when admitted to ICU, only have bacterial sepsis or more than that?

Response to reviewer: It is true that original Sweeney et al study examined only patients with bacterial infections. Here we present the first fully prospective validation of the prior study, and in a similar populationFurthermore, we here studied the endotypes in the setting a novel treatment (HAT) and so were able to begin to study a treatment enrichment effect. We have added the following paragraph to the text:

Comparing the cohorts from Sweeney et al and the current study there are some differences.   The current study was performed in community hospitals with a majority of patients being older and Caucasian with mainly with bacterial sepsis, primary bacteremia and severe pneumonia. This is largely similar to most cohorts in the prior manuscript. In addition to being the first fully prospective validation in a bacterial sepsis cohort, the present manuscript was conducted in the setting of an RCT for HAT therapy, allowing us to study the potential therapy-predictive effects of the endotypes.

It was not clear how the authors choose this subgroup (51 patients) from the ORANGES trial for the current study? How the author determined the sample size and the randomization?

Response to reviewer:  In the present study we were limited to those patients for whom a PAXgene RNA blood tube was available. This was thus a convenience subsample. As a hypothesis-generating validation study in the setting of HAT treatment, we did not predetermine sample size.  

The authors mentioned that there are 6 patients in the control (comparator) group who received open label corticosteroid therapy at the decision of the intensivist. Thus, the control group was divided into two subgroups, named control with or without corticosteroid. They further combined the HAT group with these 6 patients as corticosteroid therapy group as in Table 4B. This reviewer does not think it is appropriate since this is actual different therapy group (3 medication vs 1).

Response to reviewer We appreciate the reviewer’s insight and comments (also see reviewers comments below). As the n of the patients is small when subgroups are stratified.  We attempted Firths logistic regression in a hypothesis generating exercise.  However, the reviewer is correct.  We therefore have decided to remove the regressions and display the raw data as a new Table 4. We replace the previous paragraph describing the regressions with the followin

We sought to understand the interaction of endotype and therapy on mortality outcome. However, numbers were too small for regression testing, so we provided raw counts (Table 4). Notably, the mortality in the CE group not receiving steroids (60%) falls substantially across both groups that received steroids (29%). We emphasize that the study is small and these findings are hypothesis-generating.

In the recent report of ORANGES trial (n = 137) by the same group they authors concluded that No statistically significant change in SOFA score was found between groups, P = 0.17. No significant differences were found between study arms in ICU and hospital mortality, although they found significant difference in the time patients required vasopressors [ref 12]. However, in the current study, even in a subgroup from ORANGES, they authors found that HAT+CE can significantly eliminate the significant association of CE with mortality (Table 4A), even reduce in mortality in CE when all subjects were stratified by corticosteroid therapy (Table 4B). As the total CE patient is 12 shown in Table 2, the n of HAT within CE (HAT+CE) must be smaller. How could that reach to the above results?

Response to reviewer The reviewer is correct the n is too small to meaningfully interpret even when employing Firths regression.  After careful review we agree and have removed the regression analysis from the manuscript.  

The Table 4 has been replaced with a new Table 4 with raw numbers displaying trend. We emphasize that the study is small and these findings are hypothesis-generating. (see above)

Compare with Fig 5A and 5B, the IE survival curve seems even worse in HAT group alone vs HAT+comparator group without corticosteroids? Please confirm and discuss.

Response to reviewer We confirm and agree with the reviewer’s observation that the survival curves in the HAT therapy vs control group without steroids demonstrates small yet statistically significant decreased survival in the IE group. The survival curves of the IE in the HAT therapy alone subjects did not achieve statistical significance.  Although the n is small in these subgroups this may suggest a negative treatment effect of HAT therapy in the IE group.

We have inserted the following paragraphs:

In the KM survival curves analyzing HAT therapy and the control group subjects  not receiving corticosteroids (Fig 5A) we observed a small statistically significant decrease survival in the IE group.  This decrease in survival in the IE group was further accentuated in the KM survival curves of HAT therapy subjects alone (Fig 5B), however, this did not reach statistical significance.  Although the number of subjects is small these findings possibly suggest a negative treatment effect of HAT therapy in the IE group.

In the discussion section:

We also observed a small yet statistically significant decrease in survival of IE subjects receiving HAT therapy suggesting a possible negative treatment effect of HAT therapy in the IE group

Reviewer 2 Report

This is a very interesting manuscript.  The authors used a 33-mRNA classifier to stratify the 51 patients into 3 endotypes (IE, AE and CE) for the ORANGES trial. The results support that a 33-mRNA classifier is able to group patients into IE, AE, and CE endotypes having prognostic significance. The data will be of interest to audience in sepsis research.  

 Some comments and suggestions:

  1. More patients should be recruited for the set of experiments.
  2. The data of 33 mRNAs should be provided in a table for all the patients
  3. The endogenous levels of ascorbic acid, thiamine and hydrocortisone in patients should also be provided.

Author Response

We greatly appreciate the reviewers observations comments and recommendations.  Below are our responses.

Reviewer 2:

This is a very interesting manuscript.  The authors used a 33-mRNA classifier to stratify the 51 patients into 3 endotypes (IE, AE and CE) for the ORANGES trial. The results support that a 33-mRNA classifier is able to group patients into IE, AE, and CE endotypes having prognostic significance. The data will be of interest to audience in sepsis research. 

We agree and have added a supplemental table.

 Some comments and suggestions:

More patients should be recruited for the set of experiments.

The data of 33 mRNAs should be provided in a table for all the patients

The endogenous levels of ascorbic acid, thiamine and hydrocortisone in patients should also be provided.

We have provided a supplemental table providing the 33mRNA data (this is a large table it will be sent as an excel file )

Response to reviewer: Yes indeed we agree with the reviewer that it would be beneficial to recruit more patients, unfortunately the study is finished and therefore not enrolling any more patients. We have made clear throughout the manuscript that this is hypothesis-generating and so do not believe our conclusions are overdrawn. 

We have added ascorbic acid and thiamine levels to the tables as requested.

Round 2

Reviewer 1 Report

The authors have answered all my questions. There are still some errors in punctuation e.g. duplicates, in the manuscript.